# Fluorescence-Guided Surgery in Glioblastoma: 5-ALA, SF or Both? Differences between Fluorescent Dyes in 99 Consecutive Cases

**DOI:** 10.3390/brainsci12050555

**Published:** 2022-04-26

**Authors:** Pietro Zeppa, Raffaele De Marco, Matteo Monticelli, Armando Massara, Andrea Bianconi, Giuseppe Di Perna, Stefania Greco Crasto, Fabio Cofano, Antonio Melcarne, Michele Maria Lanotte, Diego Garbossa

**Affiliations:** 1Neurosurgery Unit, Department of Neuroscience Rita Levi Montalcini, Città della Salute e della Scienza University Hospital, University of Turin, 10126 Turin, Italy; pietro_zeppa@yahoo.it (P.Z.); armymax@hotmail.it (A.M.); andrea.bianconi@edu.unito.it (A.B.); dr.giuseppediperna@gmail.com (G.D.P.); fabio.cofano@gmail.com (F.C.).; anmelcarne@gmail.com (A.M.); michele.lanotte@unito.it (M.M.L.); diego.garbossa@unito.it (D.G.); 2Neurosurgery Unit, Department of Neuroscience and Rehabilitation, University of Ferrara, 44124 Ferrara, Italy; mmonticelli89@gmail.com; 3MRI Service RIBA Diagnostic Center, LARC, 10143 Turin, Italy; sgrecocrasto67@gmail.com; 4Humanitas Gradenigo Hospital, 10153 Turin, Italy

**Keywords:** glioblastoma, high-grade glioma, fluorescence-guided surgery, 5-ALA, sodium fluorescein

## Abstract

Background: Glioblastoma (GBM) is the most common primary brain tumor. The extent of resection (EOR) has been claimed as one of the most important prognostic factors. Fluorescent dyes aid surgeons in detecting a tumor’s borders. 5-aminolevulinic acid (5-ALA) and sodium fluorescein (SF) are the most used. Only a few studies have directly compared these two fluorophores. Methods: A single center retrospective analysis of patients treated for GBM in the period between January 2018 and January 2021 was built to find any differences in terms of EOR, Karnofsky Performance Status (KPS), and overall survival (OS) on the use of 5-ALA, SF, or both. Results: Overall, 99 patients affected by isocitrate dehydrogenase (IDH) wild-type Glioblastoma were included. 5-ALA was administered to 40 patients, SF to 44, and both to 15. No statistically significant associations were identified between the fluorophore and EOR (*p* = 0.783) or postoperative KPS (*p* = 0.270). Survival analyses did not show a selective advantage for the use of a given fluorophore (*p* = 0.184), although there appears to be an advantageous trend associated with the concomitant use of both dyes, particularly after stratification by MGMT (*p* = 0.071). Conclusions: 5-Ala and SF are equally useful in achieving gross total resection of the enhancing tumor volume. The combination of both fluorophores could lead to an OS advantage.

## 1. Introduction

Glioblastoma (GBM) or grade 4 glioma, according to the World Health Organization 2021 classification of central nervous system tumors [1], still represents the most common glioma in adults, with estimated incidence increasing by 3% per year [2,3]. Due to its malignant course, strenuous research has been performed to increase prognosis. Despite this, the median overall survival is considered to be between 12 and 15 months with maximum treatment [4], namely surgery followed by concomitant chemotherapy and radiotherapy (Stupp’s protocol) [5]. The only factor that neurosurgeons can manipulate to improve the prognosis is the extent of resection (EOR) [6]. Despite the absence of accepted and equal definition [7], gross total resection (GTR) could significantly improve the progression free survival (PFS) and overall survival (OS) of patients [8,9]. Unfortunately, GBMs behave like infiltrative tumors, and it is known that a real tumor’s extension goes far beyond its contrast-enhanced borders [10,11]. Intraoperatively, infiltrative glioma cells could easily escape from naked-eye recognition and white-light microscope. Therefore, few fluorescent agents have been investigated to perform fluorescence-guided surgery [12]: 5-aminolevulinic acid (5-ALA) and sodium fluorescein (SF) are the most explored fluorescent dyes. The former was approved by the Food and Drug Administration for glioma surgery in 2017 [13]. It is specifically metabolized by neoplastic cells; it is much more expensive, could be followed by several side effects, and requires patient-specific preparation. The latter is not specific for malignant glioma cells, but it is able to identify blood–brain barrier (BBB) impairment, much like the gadolinium of contrast-enhanced MRI; it is cheap, it does not have significant side effects, and it can be administered when a patient is already asleep on the operative table [14,15]. 

Although these are well-known differences, there are few studies that compare these two fluorophores. The purpose of this study is to investigate any differences between fluorescent dyes in terms of EOR of the gadolinium enhancing tumor volume, surgical complications (Karnofsky Performance Status, KPS), and OS in patients affected by GBM. 

## 2. Materials and Methods

A single-center retrospective analysis of patients treated for supratentorial primary brain tumor in the period between January 2018 and January 2021 was conducted. All patients with a radiological implication of high-grade glioma and a tumor location allowing for a complete resection of the contrast-enhancing area, as determined by the surgeon, were considered recruitable. Other inclusion criteria were the availability of all clinical and surgical records, preoperative and postoperative imaging, use of a fluorescent dye (5-ALA, SF or both), and histological confirmation of glioma IV grade (according to World Health Organization central nervous system tumor classification 2016). 

The exclusion criteria were: (1) a histological diagnosis other than IDH wild-type GBM; (2) a tumor that originated in the midline, basal ganglia, cerebellum, or brainstem; (3) multicentric tumors; (4) radiological findings suggesting low-grade glioma with malignant transformation; (5) medical reasons precluding MRI (for example, the presence of a pacemaker); (6) an inability to give consent because of dysphasia or language barrier; (7) a preoperative Karnofsky Performance Status (KPS) score of 60 or less; and (8) a history of active malignant tumors at any other site. 

All procedures performed for this study were in accordance with the ethical standards of “Azienda Ospedaliera Universitaria Città della Salute e della Scienza”, University Hospital, University of Turin and with the 1964 Helsinki declaration and its later amendments or comparable ethical standards. 

Regarding fluorescent dyes: 5 ALA (20 mg/kg) alone was administered orally 2.5 to 3.5 h before anesthesia induction, and SF (3 mg/kg) was administered intravenously at the induction of the anesthesia. All patients were treated with the aim of achieving a maximal safe resection of the contrast-enhancing tumor volume.

An OPMI Pentero 800, Zeiss surgical microscope equipped only with a specific filter BLUE 400 (Carl Zeiss, MeditecAG, Jena, Germany) was used until 2019. Since then, a Leica M530 OHX (Leica Microsystems, Heerbrugg, Switzerland), equipped with both FL 400 and FL 560 filters to emit and observe different wavelength ranges, was used interchangeably to detect 5-ALA and SF, respectively, and fluorescence pattern distribution. Since the introduction of the Leica operative microscope, the choice of fluorophore has been based on the preference of the primary surgeon and on the availability of dyes. In a few cases, it has been decided to employ both fluorophores to exploit the advantages of both [16].

All the procedures were conducted with the aid of a neuronavigation system (Brainlab iPlan 3.0 and Elements, BrainLAB AG, Munich, Germany or Medtronic systems Stealth Station S7, Medtronic Inc., Dublin, Ireland) and intraoperative neurophysiological monitoring. Diffusion tensor imaging and a tractography of the fascicles involved in language pathway were integrated in the neuronavigation system for tumors localized in related eloquent areas, as already described [17].

Surgical strategy contemplated a first white light inspection of the tumor, and before excision, the fluorescent pattern was systematically analyzed. During resection, especially for tumors localized in eloquent areas, a neuronavigation check was performed to realize the distance of significant white matter bundles. A thorough inspection of the margins of surgical cavity with one or both fluorescent dyes was always performed. Theoretically, all the procedures aimed for the complete resection of fluorescent tissues, unless an eloquent area or an IONM impairment limited the EOR. 

A postoperative contrast-enhanced brain MRI was performed within 48 h of the surgery. A gross total resection was considered in the absence of a contrast-enhanced tumor at postoperative MRI. A volumetric analysis was performed on preoperative and postoperative images through manual segmentation (Horos Project, www.horosproject.org), retrieved on the 13 December 2019) as described elsewhere [18].

Four groups with different amounts of EOR were identified: (1) 100% (class 1), (2) 99–90% (class 2), (3) 89–80% (class 3), and (4) less than 80% (class 4). Results were stratified for MGMT promoter status, which is a predictive biomarker of the response to adjuvant therapies [19]. One month after surgery, all patients were evaluated in a multidisciplinary setting to define the best treatment for the patient according to the current guidelines.

### Statistical Analysis

Data are expressed as mean (±standard deviation) for continuous variables and as frequencies and percentages for categorical data. Outcome variables were compared using the χ^2^ test and Fisher’s exact test for categorical variables. The comparison between continuous variables was made with a *t*-test, Mann–Whitney test and one-way ANOVA test. A Kaplan–Meier analysis and the log-rank test were used to compare OS between groups. Statistical significance level was set at *p* < 0.05. All statistical analyses were performed using Jamovi software version 2.2 (The jamovi project (2021). Retrieved from https://www.jamovi.org accessed on 20 February 2022).

## 3. Results

Ninety-nine patients affected by IDH wild-type Glioblastoma were treated in the period of the study examination (Table 1). Mean age at diagnosis was 62.3 years. 

About half of the lesions were in the left hemisphere (50.5%); in thirty patients, the tumor was located in the precentral area (30.3%), which was meant as the brain parenchyma in front the rolandic sulcus; 35 (35.4%) tumors were located in the postcentral area, and 34 (34.3%) were defined as temporo-insular GBM. The mean pre-operative KPS was 92.7/100. 

5-ALA was administered to 40 patients, SF to 44, and both fluorophores to 15. The three groups are comparable in terms of mean age, preoperative KPS, and location of lesions. 

In the study group, there were 45 (45.5%) patients with 100% of EOR, while in 36 (36.4%), an EOR of 99–90% was achieved; an EOR of 89–80% was achieved in 15 (15.2%) patients, and an EOR of <80% was achieved in 3 (3%) patients.

The mean post-operative KPS was 87.5/100. Regarding histological and molecular analysis, 51 patients (51.5%) showed an MGMT promoter methylation.

The mean overall survival was 14.9 (±9.91) months. If divided by employed fluorophore, the mean OS was 20 (±16), 12.3 (±5.7), and 18.1 (±11.9) months in 5-ALA, SF, and 5-ALA + SF groups, respectively. 

Considering different classes of EOR achieved with the use of one or both fluorophores: 18/40 of patients, in whom 5-ALA was administered, achieved 100% of EOR (class 1), 21/44 for SF and 6/15 for both; EOR class 2 was characterized by 17/40 for 5-ALA, 13/44 for SF and 6/15 for both; 4/40 for 5-ALA, 8/44 for SF, and 3/15 for both for EOR class 3; 1/40 for 5-ALA, 2/44 for SF, and 0/15 for both was the distribution of EOR for < 80% (class 4) (Figure 1). 

No statistically significant associations were identified between the fluorophore and EOR (*p* = 0.783). The results do not reach statistical significance even by aggregating the EOR classes and setting the limit to 100% (*p* = 0.872) and 90% (*p* = 0.469). Similarly, a statistical significance was not reached, either by comparing only those patients where a single fluorophore was employed (*p* = 0.523) or by stratifying by MGMT promoter status (*p* = 0.783).

Furthermore, there was not a statistically significant relationship between the employed fluorophore and the postoperative mean KPS (*p* = 0.270). In the same way, EOR was not associated with the postoperative KPS, neither aggregating for EOR of 100% or EOR of >90% (mean *p* = 0.856–0.349; median *p* = 0.903–0.309).

Survival analyses do not show a selective advantage for the use of a given fluorophore (*p* = 0.212), although there appears to be an advantageous trend associated with the concomitant use of 5-ALA and SF (Figure 2). 

This trend is accentuated after stratification by MGMT (Figure 3), although it does not reach statistical significance (*p* = 0.071).

## 4. Discussion

The use of fluorophore in glioma surgery provides a well-known advantage in terms of EOR and, consequentially, PFS and OS. The superiority of fluorescence-guided surgery (FGS) compared to white-light surgery is undebatable, as shown in different studies [20,21,22], and the possibility to achieve gross-total resection, or sometimes even a supramaximal resection, is clearly increased [23]. 

5-ALA is a natural biochemical precursor of heme. Once entering cells, it is metabolized by heme biosynthesis. In normal cells, the final product of this metabolic pathway is heme protein, while in tumor cells, protoporphyrin IX (PpIX) is the main product that accumulates inside mitochondria because of the downregulation of ferrochelatase (the enzyme that converts PpIX into heme). This intermediate product shows fluorescent properties absorbing light in the blue spectrum (375–440 nm) and emitting a red-violet light upon fluorophore relaxation. For this reason, an operating microscope with dedicated filters for excitation in the 400–410 nm wavelength range and display in the 620–710 nm wavelength range (BLUE 400 filter, Carl Zeiss Meditec AG, Jena, Germany) enables neurosurgeons to see the different degrees of “pink” or “lava orange” tissue, sharpening the perception of neoplastic area with different cell-densities [24,25] and, possibly, different molecular characteristics [26].

Indeed, 5-ALA could detect infiltrating tumor areas in newly diagnosed GBM [27,28,29]. It has been demonstrated that the presence of glioma cells even in vague 5-ALA-fluoresced areas beyond a tumor’s borders, have an improved chance of overall survival [30], although it carries a higher risk to remove non-neoplastic tissue [24,31]. Nevertheless, the opposite problem could be the impossibility to recognize eloquent areas [32], especially when the resection goes beyond the contrast-enhanced nodule, and the patient is not monitored because they are asleep. In fact, an increased rate of postoperative neurological complications has been reported in FGS compared to intraoperative stimulation brain mapping [33,34,35], questioning the role of FGS itself [36]. 

Due to a safer profile, manageable administration, and economic reasons, sodium fluorescein has recently become the preferred fluorophore in many centers. The mechanism of action is different compared to 5-ALA: it accumulates in the extracellular space in those areas where a breach arises in the blood–brain barrier (BBB). It gets excited by a light ranging from 460 to 500 nm and emits fluorescent radiation in the range 540–690 nm. Operating microscopes with blue light excitation and 540-690 nm emission filters (i.e., YELLOW 560 nm filter, Carl Zeiss, Meditec, AG) highlights the pathologic tissue from the healthy one, especially at a tumor’s borders where an increased amount of viable tumoral cells should be [37,38]. Different degrees of fluorescence intensity have been identified and correlated to tissue pathology, although they are strongly time-dependent [39]. 

Margins of a tumor where SF accumulates nearly approximate the boundaries of the contrast-enhanced nodule seen at MRI. Conversely, there have been cases of FS fluorescence in healthy brain parenchyma [40,41,42]. Since it is a specific mechanism of action, it could diffuse beyond a tumor’s boundary as a consequence of direct surgical manipulation. Some authors have advocated the use of both 5-ALA and SF to increase specificity and sensitivity for recognizing glioma cells [16]. Nevertheless, few studies have observed the presence of the fluorescent dye even in non-contrast-enhanced regions of high-grade glioma [39,43], questioning the real mechanism of action of SF. 

Most studies have focused on the advantages of FGS with 5-ALA or SF against white-light surgery. Since the relatively recent introduction of SF in glioma surgery, the results of its efficacy in obtaining an advantage in terms of EOR, PFS, and OS came mainly from retrospective case–control or non-randomized prospective studies [39,41,42,44,45,46,47]. Recently, a prospective non-randomized multicenter phase II trial showed a high sensitivity and specificity of SF to detect glioma cells, achieving a significant rate of GTR (82.6%) and 6-month PFS (56.6%).

To date, only one study has compared the two fluorophores 5-ALA and SF [48]. Although marked by the intrinsic limitations of a retrospective study and eventually by time-dependent bias, given the long period that they chose for the analysis, the authors did not find a statistically significant difference in terms of EOR, PFS, and OS. The results may sustain the equivalence between the two fluorescent dyes. Moreover, focusing on a more practical and economic view, SF could be considered superior for its own characteristics. 

Other reports in the literature on the contemporary use of both fluorophores have been in the form of small case series [16,49,50]. Among these, only Della Puppa et al., compared the two fluorescent dyes using the current accepted dosage and timing of administration. Although very small samples, they found pros and cons for both fluorophores, suggesting their eventual synergic use [50]. Considering the different time, systemic half-life, and pattern of catabolism, the contemporary use of both fluorophores does not increase the risk of photosensitive side effects.

In the present paper, the use of both fluorophores is considered an additional opportunity and is compared to the use of a single fluorophore. The absence of any statistically significant difference between the subgroups confirms the equivalence in terms of EOR and OS that Hansen et al., found in their retrospective study. Clearly, the retrospective nature of this study and the heterogeneity between subgroups (first of all, the small number of patients where both fluorophores have been used) represent important limitations of this study. However, analyzing the different classes of EOR among subgroups, we found an EOR of more than 90% in 87.5% of patients for 5-ALA, 80% of patients for both, and 77.3% for only SF. 

The contemporary use of both fluorophores shows a better trend in terms of OS. This result should be read keeping in mind the possible presence of time-dependent and selection biases. In fact, both fluorophores have been used preferentially in younger patients and when the possibility of a gross total resection had been anticipated preoperatively.

Considering the effect on gross total resection of the contrast-enhancing nodule, both fluorophores seem to achieve an equivalent result. Consequentially, the use of one fluorescent dye rather than the other should be discussed, considering the advantages and disadvantages of each one and which one is the best fitted for each patient (the tumor’s location, closeness to eloquent areas, possibility of GTR). Furthermore, economic reasons could justify the choice of SF when the gross total resection of contrast-enhancing nodule is the goal of surgery. In fact, SF costs about EUR 5 per vial, compared to about EUR 2000 for 5-ALA. Conversely, the latter, with its superiority in terms of sensitivity for infiltrated areas beyond the contrast-enhanced border of the lesion, should be used when a supramaximal resection is feasible [51]. 

## 5. Conclusions

This single-center retrospective study, including 99 consecutive cases undergoing IDH wild-type GBM resection, did not demonstrate a significant difference between the EOR of the contrast-enhancing nodule achieved by employing 5-ALA, SF, or both dyes. The same result was confirmed in the survival analysis. Despite this, a favorable trend in terms of OS was seen when combining both dyes, particularly after stratification by MGMT.

## Figures and Tables

**Figure 1 brainsci-12-00555-f001:**
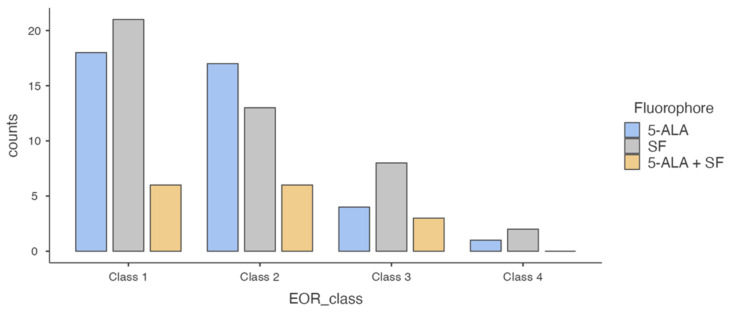
Graph of different distribution of fluorescent dyes among EOR classes. Class 1 represents 100% of resection of the contrast-enhanced nodule; 99–90% for class 2, 89–80% for class 3 and less than 80% for class 4. An EOR of more than 90% (class 1 and 2) was obtained in 35/40 patients where 5-ALA was used as fluorophore, in 34/44 for SF and 12/15 for both.

**Figure 2 brainsci-12-00555-f002:**
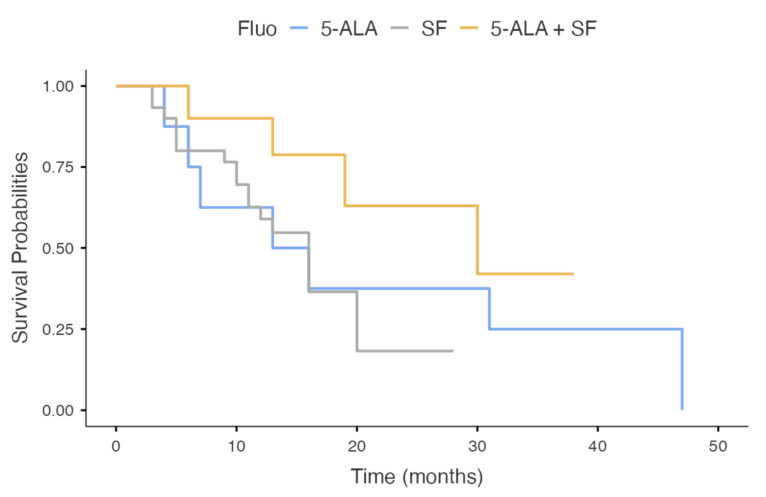
Kaplan–Meier survival analysis of OS stratified for different fluorophores. There is not a statistically significative advantage in terms of OS. A slight advantage might be present in the contemporary use of both fluorophores (yellow line).

**Figure 3 brainsci-12-00555-f003:**
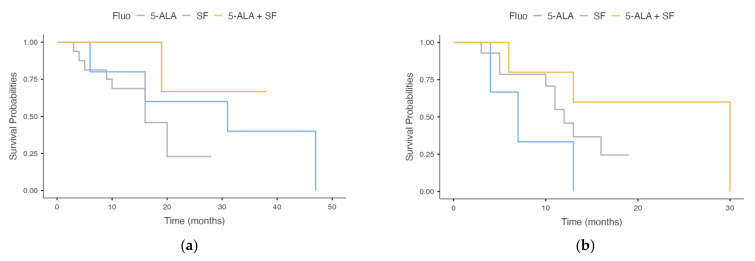
Kaplan–Meier survival analysis of OS stratified for different fluorophores and for MGMT promoter status. (**a**) MGMT promoter methylated; (**b**) MGMT promoter not methylated. The contemporary use of both fluorophores seems to provide some benefit, although statistically insignificant (*p* = 0.071).

**Table 1 brainsci-12-00555-t001:** Population characteristics. Entire population and subgroups are showed separately. Data are expressed as mean ± standard deviation for continuous variables, as frequencies and percentages for categorical data.

	Entire Population	5-ALA	SF	5-ALA + SF
	99	40	44	15
Age	62.3 ± 9.6	63.9 ± 9.5	60.8 ± 9.9	61.7 ± 8.7
Sex	F:M 50(50.5%):49(49.5%)	19(47.5%):21(52.5%)	23(52.3%):21(47.7%)	8(53%):7(47%)
Left:Right Sided	50(50.5%):49(49.5%)	22(55%):18(45%)	21(47.7%):23(52.3%)	7(47%):8(53%)
Precentral:Postcentral:Temporo-insular	30(30.3%):35(35.4%):34(34.3%)	14(35%):12(30%):14(35%)	11(25%):17(38.6%):16(36.4%)	5(33.3%):6(40%):4(26.6%)
Preoperative KPS	92.7 ± 9.9	91.9 ± 10.4	92.9 ± 10.3	95 ± 7.5
Postoperative KPS	87.5 ± 13.9	84.3 ± 16.6	90.6 ± 10.8	89.6 ± 9.4
Preoperative CE-T1w Volume	40.8 ± 29.4 cm^3^	43.9 ± 31.7 cm^3^	32.2 ± 23.2 cm^3^	48.8 ± 29.5 cm^3^
Postoperative CE-T1w Volume	0.42 ± 8.4 cm^3^	2.3 ± 11.4 cm^3^	3.3 ± 3 cm^3^	3.5 ± 0.4 cm^3^
MGMT Promoter Status	51 (51.5%) methylated	23 (57.5%) methylated	21 (47.4%) methylated	7 (47%) methylated
OS	14.9 ± 9.91 months	20 ± 16 months	12.3 ± 5.7 months	18.1 ± 11.9 months

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
