# Peer review of "Fluorescence-Guided Surgery in Glioblastoma: 5-ALA, SF or Both? Differences between Fluorescent Dyes in 99 Consecutive Cases"

_brainsci, 2022, doi:10.3390/brainsci12050555_

Round 1
Reviewer 1 Report
The presented manuscript is a one-center retrospective analysis of patients operated on for glioblastoma over a period of 3 years. The authors study the differences in terms of extent of resection Karnofsky Performance 19
status and overall survival after using 5-aminolevulinic acid, sodium fluorescein or both.The manuscript is interesting and clear. Tables and figures are legible and correctly described. There is little research in the literature to directly compare these two fluorophores. There is a factor that significantly increases the value of the article. The important conclusion is that neither of the two fluorophores is better than the other.This is an interesting article and provides new information about the fluorescent dyes used in glioblastoma surgery.
Author Response
Thank you for taking the time to review our manuscript. We appreciate the opportunity to improve quality of our work through your insightful comments.
Reviewer 2 Report
The authors conducted Fluorescence-guided surgeries (FGS) using 5-ALA and Fluorescein sodium (FS) for the treatment of WHO grade IV (2016) GBM and analyzed the results by comparing the extent of resection and survival. This is a very timely and interesting report regarding the recent advances of FGS in conjunction with the availability of multiple fluorescent dyes. Although it is a retrospective analysis of a selected cohort in a single center, there is a significant value in this type of study, particularly as a rapid report. The patient selection and the result were understandable and may help neurosurgeons make choices in performing FGS. Still, I would like to suggest some modifications to the manuscript.
- Tables
Table 1 needs to be edited with proper lines and fonts. For example, why is Age in bold? What is the line under the Age section? This will help the readers can quickly grasp the content. Also, some of the data that are only mentioned in the main text (Line 125-135) should be incorporated into Table 1. In addition, please consider rearranging the table by directly comparing the 5-ALA group, FS group, and both groups in multiple columns to show no differences in essential characteristics.
- Figures/Legends
Overall, figure legends are in short descriptions. Legends should explain the figures without referring to the main text. Please add more text to the legends. Figures 2 and 3 lack the unit on the X-axis. Days? Months? mOS of each group should be mentioned in each graph. Please clarify. Another question is: what if the authors used the Wilcoxon method instead of Log-rank to compare the survival curves?
- Discussion
In line 173, Pentero BLUE400 not only filters the observation optics but also filters the excitation light (Blue). Please refer to the manual and make this point clear, as well as the YELLOW filter. Or, do the authors use a specific (external) source of the excitation light?
It is essential to discuss the safety in comparing the two dyes, not only the cost. A particular interest is in the effect of combining the two, potentially causing an increased risk of photosensitive side effects. Did the authors have concerns about administering the two dyes simultaneously? What did the ethics committee comment, if any? Please discuss.
Another aspect of comparing the two dyes is the dynamics of reacting to the light. For example, if FS only diffuses to the tissue by the breakdown of BBB, how long does it stay in the tumor tissue? How about the photobleaching when exposed to the excitation light? Any difference between 5-ALA and FS? This type of information should be of great value to the readers.
